# Shielding and activation of a viral membrane fusion protein

Steinar Halldorsson[1], Sai Li [1], Mengqiu Li[1], Karl Harlos[1], Thomas A. Bowden [1] & Juha T. Huiskonen [1,2]

Entry of enveloped viruses relies on insertion of hydrophobic residues of the viral fusion protein into the host cell membrane. However, the intermediate conformations during fusion remain unknown. Here, we address the fusion mechanism of Rift Valley fever virus. We determine the crystal structure of the Gn glycoprotein and fit it with the Gc fusion protein into cryo-electron microscopy reconstructions of the virion. Our analysis reveals how the Gn shields the hydrophobic fusion loops of the Gc, preventing premature fusion. Electron cryotomography of virions interacting with membranes under acidic conditions reveals how the fusogenic Gc is activated upon removal of the Gn shield. Repositioning of the Gn allows extension of Gc and insertion of fusion loops in the outer leaflet of the target membrane. These data show early structural transitions that enveloped viruses undergo during host cell entry and indicate that analogous shielding mechanisms are utilized across diverse virus families.

[1] Division of Structural Biology, Wellcome Centre for Human Genetics, University of Oxford, Roosevelt Drive, Oxford OX3 7BN, UK. [2] Helsinki Institute of Life Science and Faculty of Environmental and Biological Sciences, University of Helsinki, Viikinkaari 1, Helsinki 00014, Finland. Steinar Halldorsson and Sai Li contributed equally to this work. Correspondence and requests for materials should be addressed to T.A.B. (email: thomas.bowden@strubi.ox.ac.uk) or to J.T.H. (email: juha.huiskonen@strubi.ox.ac.uk)

Despite the vast genetic and pathobiological diversity of membranous viruses, they are united by a common requirement to achieve host cell entry by fusion of their lipid bilayer envelope with a membrane of the target cell. This energetically favourable process is facilitated by viral fusion proteins, metastable molecules that fall into three architecturally distinct classes (I–III)[1]. Upon activation by acidification or receptor binding, fusion proteins in each of the three classes undergo analogous functional transitions, including the insertion of hydrophobic fusion peptides or loops into the host cell membrane, and formation of extended, hemifusion and post-fusion states. While the conformations formed before and after membrane fusion are well-characterized[1], there is a paucity of structural information with regards to the transitory conformations sampled by these proteins.

Rift Valley fever virus (RVFV; genus *phlebovirus*, family *Phenuiviridae*) is an important example of an enveloped virus that harbours a class II fusion protein. RVFV is a biomedically relevant arthropod-borne pathogen that infects both humans and livestock and is a causative agent of viral haemorrhagic fever. As with other phleboviruses, RVFV displays two glycoproteins on the lipid bilayer envelope of the virion, Gn and Gc, which form higher order pentameric and hexameric ring-shaped assemblies[2–5]. While the Gc glycoprotein is responsible for membrane fusion, the role of the Gn remains undefined. Host cell entry of RVFV is initiated following the interaction between high mannose-type N-linked glycans displayed on one or both of these glycoproteins and the C-type lectin, DC-SIGN[6,7]. Following attachment, virions are internalized by caveolae-mediated endocytosis[8] and transported to late endosomal compartments[9]. Virus entry is completed upon Gc-mediated fusion of the viral envelope and host cell membrane[10], in a pH-dependent mechanism[11–13].

Our current understanding of phlebovirus envelope structure is limited to low-resolution maps of entire virions[2–4,14] and high-resolution crystal structures of the Gc, a class II viral fusion protein, in pre-fusion[11] and post-fusion conformations[12]. The precise organisation and specificity of Gn–Gc spike interactions have remained elusive, especially given the unknown fold architecture of the Gn. Detailed understanding of these interactions is required to understand how the fusogenic potential of the metastable Gc is arrested at neutral pH and activated upon exposure to the acidic environments of endosomal compartments.

Here, we sought to address the organization of Gn–Gc spike complex, as displayed on the mature virion, and the conformational rearrangements these proteins undergo during the early stages of membrane fusion. We determined the high-resolution crystal structure of the RVFV Gn ectodomain N-terminal region and revealed a unique fold organization reminiscent to that of the Gn of hantaviruses and E2 of alphaviruses. Integration of our RVFV Gn with that of a previously reported RVFV Gc crystal structure into an electron cryo-microscopy reconstruction of the entire RVFV virion clarified the organization of the mature heteromeric Gn–Gc glycoprotein spike complex, revealing that the Gn shields the hydrophobic fusion loops of the Gc. Tomography analysis of acidified virions in the presence of liposomes, a surrogate for a host cell membrane, demonstrated how the metastable Gn–Gc complex rearranges to allow insertion of Gc-resident fusion loops into the outer leaflet of the host membrane. These data provide a molecular-level framework for understanding the early stages of glycoprotein-mediated fusion.

## Results

**Gn glycoprotein crystal structure reveals a novel fold**. We solved the crystal structure of RVFV Gn (Fig. 1a) to 1.6-Å resolution (Supplementary Table 1). Although, residues 154–560

were present in the RVFV Gn construct, electron density corresponding to only residues 154–469 was observed, indicating that the remaining 91 C-terminal residues were likely cleaved during

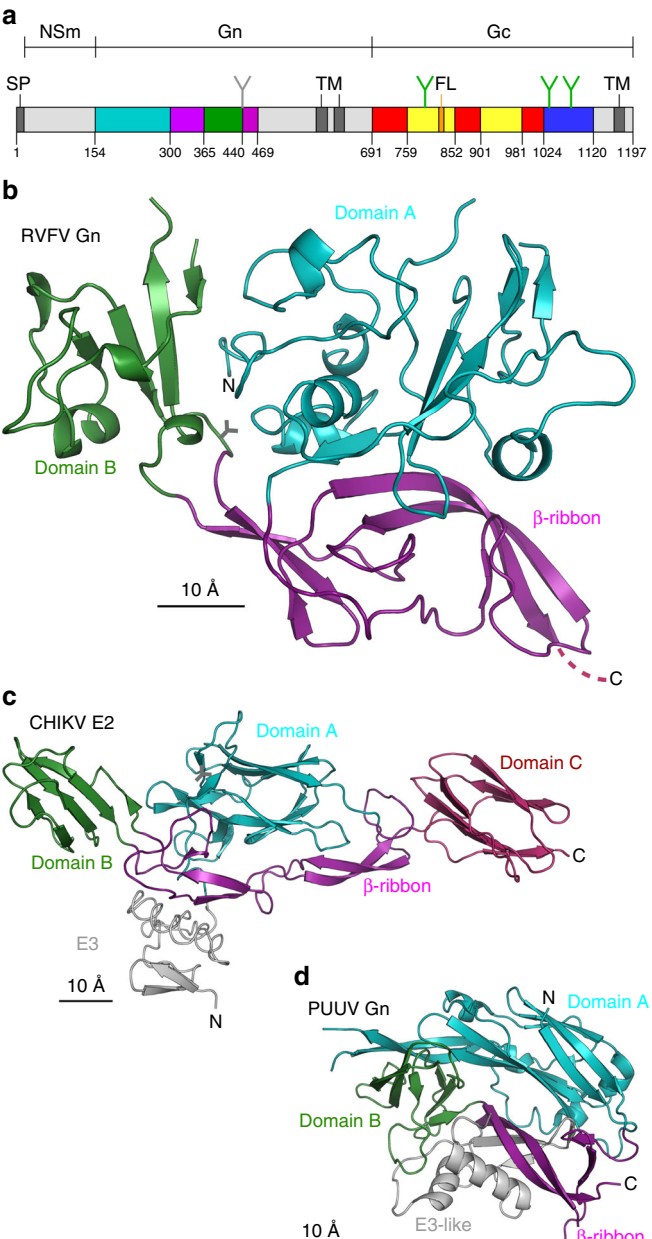

**Fig. 1** Novel fold of Rift Valley fever virus Gn glycoprotein. **a** The domain organisation of the M-segment of RVFV showing the domain organisation of Gn and Gc by colour. Domain A is teal, domain B is forest green, the β-ribbon is purple, domain l is red, domain II is yellow, domain III is blue and the hydrophobic fusion loop (FL) is orange. Transmembrane regions are indicated with TM, signal peptide is indicated with SP and glycan positions are indicated with a Y. **b** The crystal structure of Gn from Rift Valley fever virus. Domains are coloured as in **a** and the position of a predicted glycan is indicated with a Y. **c** Structures of E2 and E3 from Chikungunya virus (CHIKV)[17]. **d** Structure of Gn from Puumala virus (PUUV)[15]. RVFV Gn adopts a novel fold but shares a similar domain organisation with CHIKV E2. Both proteins have a β-ribbon with two anti-parallel β-sheets and a domain B inserted into the β-ribbon. In CHIKV, E2 domain B is a β sandwich, while in RVFV Gn domain B is a small β sheet decorated with four short helices. Domain A of CHIKV E2 and PUUV Gn are both a seven stranded β sandwich with the same topology (Supplementary Figure 1)[61]

crystallogenesis. RVFV Gn adopts a novel fold comprising 15 β-strands, 5 α-helices and 4 $3_{10}$-helices (Fig. 1b). The protein can be divided into three domains: a predominantly helical N-terminal domain termed 'domain A' (residues 154–305), a β-ribbon domain (residues 306–365 and 440–469) and a small globular domain termed 'B' (residues 366–439). RVFV Gn exhibits a similar domain organisation to both hantaviruses[15,16] and alphaviruses[17] (Fig. 1c). Although structurally distinct, RVFV Gn exhibits a greater level of secondary structure conservation with the alphaviral E2 than with the hantaviral Gn (Fig. 1d; Supplementary Figure 1), indicating a closer than expected evolutionary relationship between the phleboviral Gn and alphaviral E2.

**The flexible Gn–Gc glycoprotein surface of the virion**. We sought to understand the higher order assembly of the Gn–Gc spike complex in the context of the whole RVFV virion. The earlier electron cryo-microscopy (cryo-EM)-based single-particle reconstructions of the virion have been limited to resolutions worse than 20 Å, presumably due to inherent flexibility of the particles[2,4]. Our tomography characterization of purified native virions revealed that while virions were uniform in size (~100 nm in diameter), they often deviated from spherical shape (Supplementary Figure 2a, b). In our attempts to preserve the shape and icosahedral symmetry during purification, we devised an improved sample preparation strategy based on fixation of the particles with formaldehyde by dialysis, followed by sucrose gradient purification (Methods; Supplementary Fig. 2c, d). Single-particle averaging and 3D reconstruction (Supplementary Table 2) yielded a 13-Å resolution reconstruction of the Gn–Gc spike envelope (Fig. 2a, b; Supplementary Fig. 2g). As previously shown, the glycoprotein shell is composed of 720 Gn–Gc heterodimers following $T = 12$ icosahedral organisation[2,4], which encapsulates the tri-segmented genome and intraviral proteins.

The resolution of the reconstruction was most likely limited to 13-Å due to remaining flexibility of the particles, despite efforts to include in the reconstruction only the most ordered particles by manual selection and 3D classification (Supplementary Fig. 2e, f). The resolution was further improved by application of our localized reconstruction method[18], which accounted for the flexibility remaining in the fixed RVFV virions. Localized reconstruction of each of the four types of the Gn–Gc capsomers (pentamers and hexamers of type 1–3) resulted in four independent structures at resolutions between 7.7 and 8.6-Å (Supplementary Figure 3, 4; Supplementary Table 3). Imposing lower than six-fold symmetry (two-fold symmetry for the type 2 hexamer and three-fold symmetry for the type 3 hexamer), or no symmetry (type 1 hexamer) gave the highest resolution, suggesting significant deviations from six-fold symmetry in the hexamers (Supplementary Figure 3). Local resolution analysis revealed that membrane-proximal regions of the capsomers were the best resolved (Supplementary Figure 4), suggestive that higher degrees of flexibility may exist in the membrane-distal regions of the virus. The localized reconstruction approach was further validated by comparison to structures resolved by sub-tomogram averaging, in addition to the conventional icosahedral reconstruction (Supplementary Figure 5).

**Gn shields the Gc fusion loops in the pre-fusion state**. To determine the location of RVFV Gn and Gc subunits on the RVFV virion surface, we adopted a molecular dynamics flexible fitting approach[19] (Supplementary Figure 6; Supplementary Movie 1). We first performed fitting with the localized reconstruction of the pentamer at 7.7 Å resolution, followed by fitting the resulting Gn–Gc heterodimer model to localized reconstructions of the hexamers at 8.0–8.6 Å resolution. This sequential

fitting approach allowed us to create a full atomic description of the Gn–Gc glycoprotein shell by applying icosahedral symmetry to the 12 Gn–Gc dimers in the asymmetric unit (Methods; Fig. 2c). Consistent with previous biochemical and structural studies[2,20,21], this fitting verified that the Gn and Gc assemble as heterodimers, which form higher order ring-like pentameric and hexameric capsomer structures on the mature virion (Fig. 2c, d, e). Domain A of the Gn forms a ring on top of these capsomers interacting with domain II of Gc, occluding a surface area of ~950 Å$^2$ (Fig. 2d). Higher resolution reconstructions, however, are needed to define the exact nature of this interaction. As the N-linked glycosylation displayed on viruses is unlikely to be obscured in protein–protein interfaces[22,23], mapping of putative N-linked glycosylation sites onto pentameric and hexameric Gn–Gc capsomers was performed to validate the model. This analysis revealed all putative N-linked glycosylation sites to be either solvent exposed (RVFV Gn Asn438 and RVFV Gc Asn794, Asn1035) or facing unoccupied density that could partially correspond to an ordered glycan (Gc site Asn1077; Supplementary Figure 7).

Surprisingly, unlike the rod-like array of extended anti-parallel Gc dimers predicted from the low-resolution modelling attempts[11], we observed that RVFV Gc forms a kinked conformation (Fig. 2e), not previously sampled in crystallographic investigations of class II fusion glycoproteins (Supplementary Figure 8). While such degree of conformational plasticity is unprecedented in class II fusion proteins, we would like to note that, albeit to a lesser extent, plasticity in the same region has been observed to be necessary for the icosahedral assembly of the fusion glycoprotein in an alphavirus[24].

The Gn–Gc interface is formed between the β-ribbon domain of the Gn and the hydrophobic fusion loops of Gc[11], which are shielded from solvent at the interface between Gn domains A and B (Fig. 2f). This is reminiscent to alphaviruses SFV and CHIKV, where the fusion loop of the class II fusion protein E1 protein is shielded by the E2 partner protein (Fig. 2f)[17,24,25] but contrasts the homotypic shielding observed in flaviviruses such as ZIKV[26] and DENV[10,27].

**Membrane insertion of the fusion loops**. We sought to understand the metastable nature of the Gn–Gc capsomer and the structural transitions that the complex undergoes during Gc-mediated fusion. Consistent with our previous studies on the related phlebovirus, Uukuniemi virus (UUKV)[13], fluorescence spectroscopy analysis confirmed that the endogenous lipid, bis (monooleoylglycero)phosphate (BMP) is a required factor for viral fusion at acidic pH. Lipid mixing index with BMP was 0.26 ± 0.02 (standard error of the mean, $N = 6$) and without BMP 0.04 ± 0.01 (standard error of the mean, $N = 6$; $P < 0.01$; two-tailed two-sample t-test assuming equal variances).

We cryogenically trapped RVFV in the presence of liposomes containing BMP at acidic pH and performed cryo-electron tomography on these RVFV–endosome mimicking complexes (Fig. 3a, b; Supplementary Table 3). Sub-tomogram averaging (Supplementary Table 4; Supplementary Figures 9, 10) and MDFF fitting (Fig. 3c) revealed that while non-membrane-facing capsomers retain the pre-fusion configuration, membrane-interacting pentamers undergo pronounced conformational rearrangements that allow the insertion of the Gc-residue hydrophobic fusion loops into the target membrane. This insertion event is achieved, in part, by a dramatic translational shift of the Gn, which unshields the Gc allowing it to form an extended intermediate state. (Fig. 3d; Supplementary Figure 11). The depth of insertion predicted by this molecular dynamics analysis agrees with the hypothesis that the main chains of the

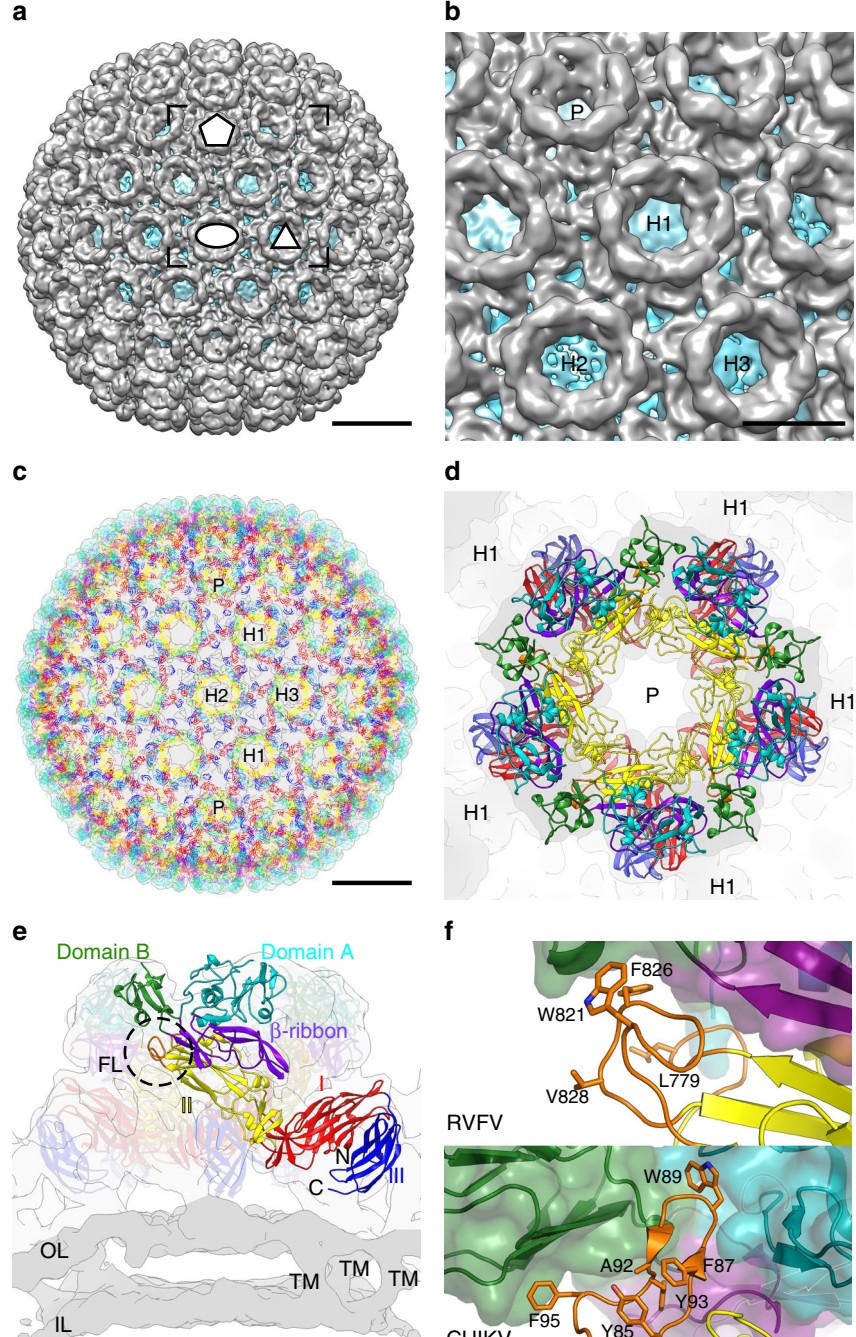

**Fig. 2** Shielding of the Gc fusion loop in the Gn–Gc glycoprotein shell. **a** An icosahedral reconstruction of the virion at 13.3-Å resolution is shown. An icosahedral two-fold (ellipse), three-fold (triangle) and five-fold (pentagon) axis of symmetry are marked. Scale bar, 20 nm. **b** A close-up of the area marked in **a**. A pentamer (P), type 1 hexamer (H1), type 2 hexamer (H2) and type 3 hexamer (H2) is labelled. Scale bar, 10 nm. **c** Virion model is shown along the icosahedral two-fold axis of symmetry. Atomic models of the pentamers and hexamers were fitted to the 13-Å icosahedral reconstruction (transparent surface). One pentameric Gn–Gc capsomer (P) and three types of hexameric Gn–Gc capsomers (H1, H2 and H3) are labelled. **d** Pentameric capsomer is shown from the top. Localized reconstruction of the corresponding density at 7.7-Å resolution is shown as a transparent surface. Due to limited resolution of the localized reconstruction, secondary structure elements were constrained in MDFF fitting to those observed in the crystal structures. We cannot, however, exclude the possibility that differences exist at secondary structure level between the native and crystallographic conformations. **e** Pentameric capsomer and the corresponding density is shown from the side. One Gn–Gc heterodimer is highlighted. Different domains and N-terminus and C-terminus are labelled. The fusion loop (FL) is circled. Outer (OL) and inner (IL) leaflets of the lipid bilayer are coloured in darker shade of grey and examples of transmembrane densities (TM) are labelled. Scale bar, 20 nm. **f** Close up of the Gn–Gc interaction (top) showing the shielding of the RVFV fusion peptide and reminiscent shielding of the fusion peptide in Chikungunya virus (CHIKV; bottom). Gn domains are coloured as in Fig. 1. Gc domain I is red, domain II yellow and domain III blue. Fusion loops are orange

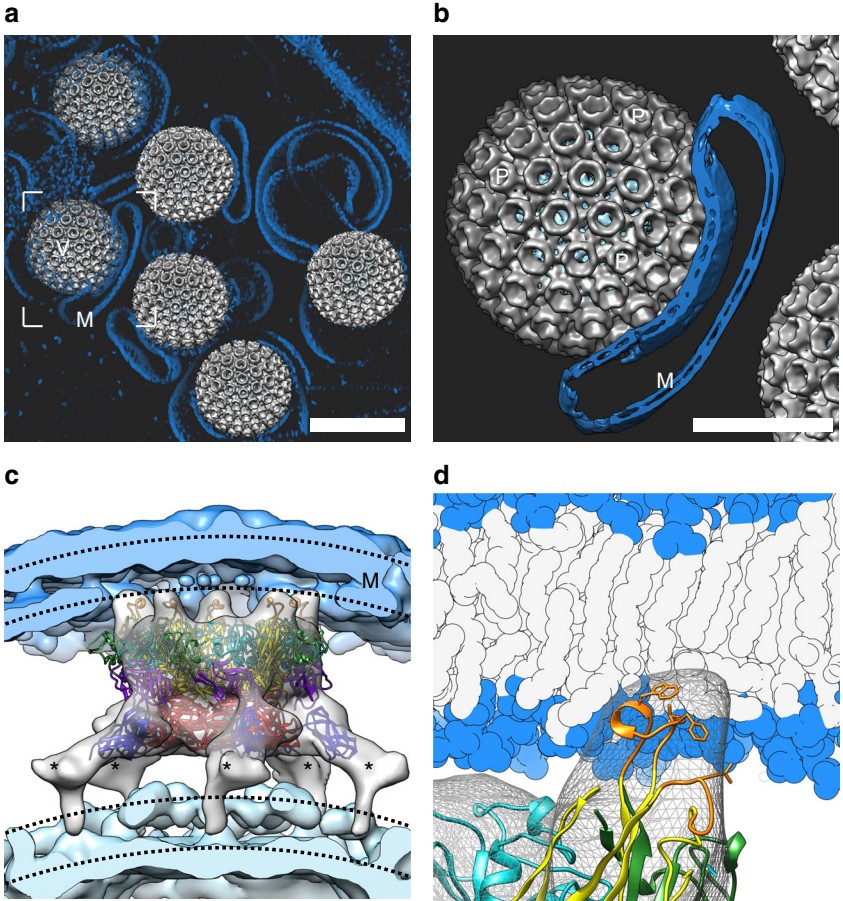

**Fig. 3** Gc fusion loop unshielding and target membrane insertion. **a** A volumetric rendering is shown for tomographic reconstruction of purified RVFV virions and liposomes (M) incubated for ~30 s at pH 5.0. Surfaces of RVFV virions (V) were rendered by placing sub-tomogram averages of the virions back in the density. Scale bar, 100 nm. **b** A close-up of one virion sub-tomogram average with pentameric capsomers labelled (P) and showing a close interaction with the liposome membrane (M). The liposome membrane was rendered by placing sub-tomogram averages of liposome membrane back in the tomogram. Scale bar, 50 nm. **c** Sub-tomogram average of the pentamer at pH 5.0 with its apical densities embedded in the target liposome membrane (blue). Fitted X-ray structures of Gn and Gc are represented as ribbons and coloured as in Fig. 2. The fusion loop (FL) is in orange. Gn–Gc capsomers are rendered as a grey surface, viral membrane is in teal and liposome membranes are in light blue in **a–c**. The locations of the two leaflets in both the viral and the liposomal membrane are indicated with a dashed line. Unoccupied densities, assigned to C-terminal parts of the Gc and Gn glycoproteins that are absent from the fitted crystal structures, are indicated with asterisks. **d** Close-up of the inserted fusion loop. The liposome membrane is shown as a cartoon with lipid head groups in blue and hydrophobic tails in grey. The sub-tomogram average is shown as a mesh

fusion loops interact tightly with lipid heads, while the aromatic side chains are projected into the hydrophobic region of the lipid bilayer[28]. Interestingly, these capsomer–membrane contacts were unique to the pentamers facing the target membrane at the early stage of membrane fusion, and were not observed in the hexamers or a pH-neutral control sample (Supplementary Figure 9).

## Discussion

These results show that RVFV Gc undergoes pH-dependent rearrangements and provide a model for the first steps of virus–host cell fusion (Fig. 4; Supplementary Figure 11). Although the low pH sensing mechanisms that drive the observed changes remain unknown, histidines in the Gc have been suggested to play a role in low pH sensing[12,29]. Following insertion of the fusion loop, non-covalently associated Gn–Gc dimers are expected to dissociate, leaving extended Gc monomers free. Such monomers would be consistent with the bridge-like densities observed in our cryo-EM analysis (Supplementary Figure 9) and have been similarly observed in other viruses harbouring class II fusion

proteins, such as UUKV[13] and Sindbis virus[30] (an alphavirus). The formation of stable post-fusion trimers, observed by crystallography[12], is likely to follow full membrane merger.

Recently, Wu et al.[31] reported the crystal structure of the RVFV Gn glycoprotein ectodomain N-terminal fragment. Superposition analysis revealed a high level of structural similarity between that Gn structure and the one reported in this study (0.4 Å root-mean square deviation between 261 pairs of C-alpha atoms, respectively). In light of the preserved nature of the RVFV Gn fold architecture with the Gn of another phlebovirus, severe fever with thrombocytopenia syndrome virus[31], we propose that our model of RVFV Gn–Gc hetero-dimerisation and higher order assembly (Fig. 2c–e) is likely to be observed across the viruses within the *phlebovirus* genus. Furthermore, our work also clarifies earlier hypotheses on the placement of the Gn[31] and Gc[11] in the RVFV virion. Indeed, the improved resolution of the density maps for the capsomers reported here (7.7–8.6 Å), combined with high-resolution crystal structures, allowed the construction of an accurate atomic model of the whole virion surface (Fig. 2; Supplementary Movie 1).

Our model of the early RVFV fusion intermediate shows that the fusion loops of the Gc glycoprotein embed in the host

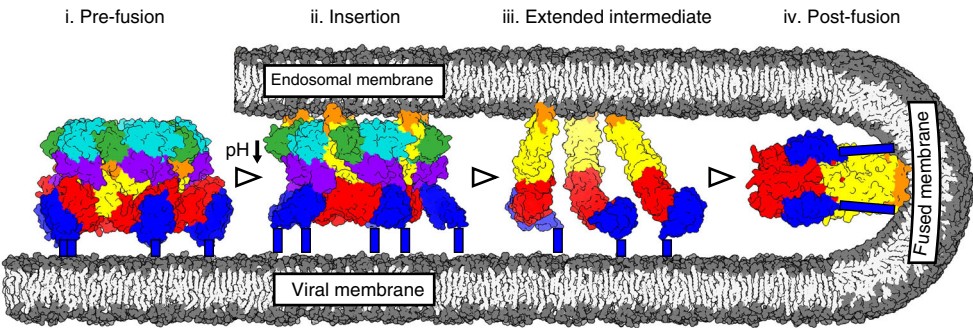

**Fig. 4** Model for phlebovirus Gc-mediated membrane fusion. (i) In the pre-fusion state at neutral pH the Gc fusion loops (orange) are buried in the structure. (ii) At the fusion permissive pH below 5.0 (downward arrow)[9] and upon exposure to a target membrane, the Gn moves to the side of the Gc, fully exposing the fusion loops at the tip of the Gc domain II. Consequently, full extension of Gc leads to the insertion of the fusion loops in the target membrane. (iii) The Gn–Gc dimers dissociate, allowing extended Gc monomers to redistribute on the viral membrane. (iv) After the merging of the endosomal and viral membranes, Gc adopts a trimeric post-fusion conformation (modelled in SWISS-MODEL[62] using a phleboviral Gc post-fusion trimer PDB:5G47 as a template) where fusion loops are embedded in the fused membrane. Gn and Gc domains are coloured as in Fig. 2 and membrane is coloured in grey, with lipid head groups in darker and acyl chains in lighter shade. C-terminal part of Gc domain III that links the protein to the viral membrane is missing in the crystal structures and is indicated with blue rectangles

membrane at low pH. The accuracy of the fitted model, however, is limited by the relatively low resolution of the sub-tomogram average of the membrane facing pentamer (20 Å). In general, rigid-body fitting to maps to comparable resolutions can reach an accuracy as high as ~4 Å[32]. In our case, the accuracy is expected to be lower, due to the fact that flexible fitting was required. Thus, the molecular details of the major conformational change in the Gn–Gc complex remain to be addressed in further studies. Interestingly, the embedding of the fusion loops was only observed in the case of the pentamers and not with the hexamers. Although it is tempting to speculate that the pentamers might have a lower activation barried to escape their metastable state than the hexamers, possibly due to less favourable subunit–subunit angles (72° in the 60 heterodimers making the pentamers as opposed to ~60° in the 660 heterodimers making the hexamers), this hypothesis remains to be tested. Furthermore, by our sub-tomogram averaging approach, we cannot exclude the possibility that a small population of the hexamers is also engaged with the target membrane, or conversely, that a small population of the pentamers facing the membrane is still in a pre-fusion state.

In toto, our integrative structural biology approach reveals the shielding of the Gc fusion loop in the pre-fusion state by its partner protein Gn. The shielding mechanism is analogous to alphaviruses, suggesting that this mechanism is widely utilized amongst enveloped viruses. The conserved presence of these features is suggestive that class II fusion glycoproteins undergo similar conformational rearrangements during host cell entry. The observed Gc unshielding at late endosomal pH and target membrane insertion constitute the first characterized early fusion intermediates. How Gc proteins merge the target membrane into the viral membrane, the number of Gc monomers that are involved in merging the membranes, and how they assume a trimeric conformation in the fused membrane[1,10,12] remains to be addressed by structural analysis of later intermediates. Preventing the conformational changes necessary for fusion loop unshielding by small molecule fusion inhibitors or therapeutic antibodies may provide a viable antiviral strategy to treat infections of RVFV and other emerging viruses utilizing this mechanism.

## Methods
**Protein expression and X-ray crystallography.** Residues 154–560 from the M-segment of RVFV (UniProt accession no. P21401, synthesized by GeneArt, Germany) were cloned into the pHLSec vector with a hexa-histidine tag[33]. The Gn construct was expressed in HEK293 cells (ATCC CRL-1573) by transfecting with 2

mg of DNA per liter of cell supernatant in the presence of 1 mg/L of kifunensine to limit glycan content. Supernatant was collected after 4–5 days post-transfection and dialyzed against buffer (10 mM Tris-HCl pH 8.0 and 150 mM NaCl) after clarification. Protein was captured by nickel immobilized metal affinity chromatography with a HisTrap column and isolated by size exclusion chromatography with a Superdex 200 10/30 column. The protein was concentrated to ~8 mg/mL and crystallised using the sitting-drop vapour method[34] at room temperature. Crystals appeared between 238 and 291 days in a solution containing 20% w/v PEG 6000 and 100 mM HEPES pH 7.0 and were cryo-cooled in the precipitant solution with additional 25% glycerol. A subset of crystals were soaked in precipitant solution saturated with $K_2PtCl_4$ for 70 min prior to cryo-cooling for phasing. X-ray data was collected at beamline I03 at Diamond Light Source (UK) on a PILATUS 6 M detector at a wavelength of 0.97625 Å and 1.07146 Å for native data and Pt-derived data, respectively. X-ray data was indexed, integrated and scaled in XIA2[35]. Phases were obtained from Pt-derived data using the single-wavelength anomalous dispersion method in autoSHARP[36] and Buccaneer, as implemented in auto-SHARP, was used to build an initial model[37]. Refinement was carried out in PHENIX[38] using translation–libration–screw-rotation restraints. Coot was used for manual model building[39]. The final model was validated by MolProbity[40]. Supplementary Table 1 contains crystallographic data collection and refinement statistics. Supplementary Figure 12 shows a stereo image for a portion of the electron density map.

**Propagation and purification of Rift Valley fever Clone 13 particles.** RVFV Clone 13[41] was provided by Friedemann Weber (Philipps University in Marburg, Germany) and handled at containment level 3 laboratory of the Oxford Particle Imaging Centre, University of Oxford, UK. Vero cells (ATCC CCL-81) were maintained in Dulbecco's modified Eagle's medium supplemented with L-glutamine, non-essential amino acids and 10% v/v fetal bovine serum. Cells were infected at an m.o.i. of 0.1 in serum free media and ~48 h post-infection supernatant was collected and clarified. Virus was isolated from supernatant by pelleting through a 20% w/v sucrose cushion in 20 mM Tris pH 7.4 and 100 mM NaCl (T20N100) and resuspended overnight in T20N100 buffer.

To better preserve the virion ultrastructure for single-particle cryo-EM, the supernatant was collected from the infected cells as above. The virus particles were chemically fixed by dialysing the supernatant against PBS containing 0.2% v/v formaldehyde for ~20 h. Excess formaldehyde was removed by dialysing against PBS. The fixed virus was concentrated by reverse dialysis using solid 35 kDa PEG and purified by gradient ultracentrifugation on a 20–60 % w/v sucrose gradient in T20N100 buffer. Gradient fractions were harvested using a gradient fractionator.

**Liposome preparation and virion–liposome fusion assays.** Lipids (Avanti Polar Lipids, AL, US) used in this study were: 1,2-dioleoyl-sn-glycero-3-phosphocholine (DOPC), 1,2-dioleoyl-sn-glycero-3-phosphoethanolamine (DOPE), bis(mono-oleoylglycero)phosphate (S,R isomer; BMP) and cholesterol (ovine; Chol). The lipid composition of liposomes containing BMP was DOPC:DOPE:BMP:Chol = 3.6:1.4:3.0:2.0 (molar ratio). For liposomes without BMP the composition was DOPC:DOPE:Chol = 3.6:1.4:2.0 (molar ratio). Lipids were dissolved in HPLC-grade chloroform (Fisher Scientific, UK) and mixed in glass vials. The chloroform was carefully evaporated under a stream of nitrogen gas and residual solvent was removed by placing the vials in a vacuum desiccator (~300 mBar) for 1–2 h. The dried lipid mixture was hydrated in buffer (6 mM succinate, 22 mM sodium phosphate, 22 mM glycine, 100 mM NaCl, pH 7.5; SPG50N100) at 5 mg/mL. The

vial was vortexed in the SPG50N100 buffer to resuspend the lipids in the buffer and three freeze–thaw cycles were applied to the suspension. Liposomes were extruded through a 400-nm track-etched polycarbonate membrane (GE Healthcare, UK), stored at 4°C and used within a week.

For each membrane fusion experiment, 100 μL RVFV (~2 × 10⁸ PFU/mL) were stained in 1 μM lipophilic DiD fluorescent dye (Thermo Fisher Scientific, UK) for ~1 h. DiD stock solution was a 1 mM ethanol solution, therefore, the concentration of ethanol was 0.1% (v/v) in the staining solution. Unbound DiD was removed in a 40-kDa size exclusion column (Zeba; Thermo Fisher Scientific). This procedure stained RVFV with DiD to a self-quenching concentration and fusion with target membrane (liposomes) resulted in dequenching and increased fluorescence emission. For each membrane fusion experiment, a 2 mL mixture containing 100 μL DiD-stained RVFV and 30 μL liposomes (final lipid concentration 0.1 mM) was prepared in a sealed quartz cuvette (optical pathlength 1 cm) in a biosafety cabinet. The cuvette was sealed and disinfected prior to the fluorescence spectroscopic measurement. The change in the DiD fluorescence was recorded first at pH 7.5 for about 300 s (Cary Eclipse fluorescence spectrometer; Agilent Technologies, UK). The mixture was acidified to pH 5.0 by adding a precalibrated amount of 1 M HCl and the change in the DiD fluorescence was recorded for about 10 min. To determine the fully de-quenched DiD fluorescence intensity, 1% Triton X-100 (Sigma-Aldrich, UK) was added to disrupt the viral envelope and the liposomes. The cuvette was sealed and disinfected. DiD fluorescence was recorded until equilibrium was reached. The cuvette was then moved back the biosafety cabinet for confirming the pH. The temperature was maintained at 37°C during the spectroscopic measurement. The excitation wavelength was 650 nm (5 nm slit width), and the emission wavelength was 665 nm (5 nm slit width). Signal integration time was 1 s, and the fluorescence intensity was sampled at 1 Hz. Lipid mixing index was defined as the ratio of fusion-induced change in fluorescence intensity to detergent-induced change in fluorescence intensity. The time-trace before acidification was fitted to a straight line ($F_1$). The time-traces after acidification ($F_2$) was fitted to a double exponential decay function. The time-traces after the addition of Triton X-100 ($F_3$) were fitted to a single exponential decay function. The fluorescence intensities were corrected for the volume change caused by acidification and the addition of detergent. The fluorescence intensities were extrapolated to the end of the recorded data using the fitted parameters. Lipid mixing indices were calculated at the endpoint ($(F_2-F_1)/(F_3-F_1)$). Data processing was done in MATLAB R2016a (MathWorks, MA, USA). Data were collected from two batches of separate virus production, each batch providing half of the data, and each experiment was conducted a total of six times.

**Electron cryo-microscopy of fixed and native virions and virion–liposome complexes.** For samples of fixed and native virions, a 3-μL aliquot was applied to a glow-discharged copper grid coated with holey carbon (C-flat 2/2; Protochips, Raleigh, NC). For tomography, a 3-μL aliquot of 6-nm colloidal gold beads (Aurion, The Netherlands) was added. Grids with virion–liposome complexes were prepared by adding an 8-μL aliquot of a 1:1:2 mixture of virus:liposomes:gold beads where the final concentration of liposomes is ~1.3 mg/mL. To trigger fusion, the grid was floated over 100-μL of SPG50N100 pH 5.0 buffer at 37° C for ~3 s prior to vitrification approximately 30 s after acidification. Grids were vitrified by plunge-freezing into ethane/propane liquid mixture using a vitrification apparatus (CryoPlunge 3; Gatan, Pleasanton, CA). Data were acquired using a 300-kV transmission electron microscope (Tecnai F30 'Polara'; FEI, Eindhoven, the Netherlands) equipped with an energy filter (slit width 20 eV; GIF Quantum LS, Gatan) and a direct electron detector (K2 Summit, Gatan). Data were collected at electron counting mode at calibrated magnification of 37,037× corresponding to pixel size of 1.35 Å at specimen level. For single-particle analysis, movies (88 frames, each frame 0.2 s) were collected at dose rate of 2.5 e⁻ per pixel/s, resulting in a total exposure of 22 e⁻ per Å². For tomography, the stage was tilted from −30 to +60° at 3 degree steps and a movie was collected at each tilt (eight frames, each frame 0.4 s) at exposure rate of 2.8 e⁻ per pixel/s, resulting in a total exposure of 150 e⁻ per Å². Data collection parameters are summarised in Supplementary Tables 2 and 3.

**Single-particle data processing and model refinement.** Movie frames were aligned using MotionCorr[42] and contrast transfer function (CTF) was estimated with CTFFIND3[43]. A total of 4,336 virus particles that showed no evident distortions were extracted from 943 averaged movies using box size of 1024 × 1024 pixels. Initial 2D and 3D classifications were performed with particles binned by a factor of four to speed up calculations in Relion 1.4[44]. A previous structure of RVFV (EMD-1550) was used as an initial model in 3D classification. To extract an ordered subset of particles for final refinement, particles were binned by a factor of two (box size of 512 × 512 pixels with pixel size of 2.7 Å) and subjected to 3D classification to 10 classes. No further alignment was performed in this final round of classification to speed up the computations. A subset of 2,995 particles was selected from the three most ordered classes (Supplementary Fig. 2e) and refined using particles gold-standard protocols in Relion to produce the final model. A mask defining the glycoprotein shell was created and FSC was calculated within the mask using relion_postprocess. Reconstruction statistics are listed in Supplementary Table 2.

**Localized reconstruction.** The localized reconstruction method (http://github.com/OPIC-Oxford/localrec)[18] in conjunction with partial signal subtraction to remove all but one capsomer density from the images[45,46] was used to calculate separate density maps of the pentamer and type 1–3 hexamers. First, a vector defining the location of each type capsomer was defined using the reconstruction of the full virion in UCSF Chimera[47]. A soft-edged spherical mask was generated to define the capsomer boundaries and used to subtract density corresponding to other capsomers, membrane and genome. Sub-particles, corresponding to the projections of individual capsomers in the subtracted particle images were extracted. The extracted sub-particles for each capsomer type were processed separately in Relion 1.4 using standard single-particle, gold-standard protocols[44]. A starting model was generated for each capsomer type with relion_reconstruct and the sub-particles were subjected to 3D classification. Sub-particles from the highest resolution 3D classes were selected for further 3D refinement. Maps were post-processed as above. Reconstruction statistics are listed in Supplementary Table 2.

**Tomogram reconstruction and sub-tomogram averaging.** Movies were corrected for drift in MotionCor2[48] by averaging eight frames for each tilt. Tilt series were dose-weighted by taking into account the accumulated dose[49]. Unweighted, raw tilt series were used to estimate the defocus parameters using Gctf[50]. The estimated defocus was used to correct the CTF of the dose-weighted tilt series by ctfphaseflip in IMOD using strip width of 20 pixels[51]. Tomograms were reconstructed by IMOD[51], managed in Dynamo Catalogue[52] and binned by a factor of 2 resulting in the final pixel size of 2.7 Å/pixel. A total of 120 RVFV virions at pH 7.5 and 30 virions at pH 5.0 were picked and extracted into boxes of 540 × 540 × 540 voxels for further analysis. Sub-tomogram averaging was carried out in Dynamo, following protocols we have established earlier[15,53]. In the first stage, virion tomograms were aligned using a previously published structure (EMD-1550) as the template. The resolution was restricted to 40 Å, and icosahedral symmetry and further binning by factor of 2 was applied at this stage. In the second stage, locations of capsomers were calculated and 12 pentameric and 110 hexameric capsomers were extracted into boxes of 128 × 128 × 128 voxels from the raw tomograms using the sub-boxing method[54]. A customized 'gold-standard' refinement method[53] was used to align the capsomers. Type 1, 2 and 3 hexameric capsomers were first aligned and averaged independently and then combined into one hexamer map. The resolution of the maps was estimated by FSC in Dynamo. Reconstruction statistics are listed in Supplementary Table 4.

To separate capsomers that were in close proximity to the target membrane from those that were not, viral membrane surfaces that were facing the target membranes were manually selected in Dynamo using tomoview[52]. Capsomers within a distance of 6 nm to the selected surfaces were processed separately from the rest. Averages of capsomers in both sets were used as templates for multireference alignments, by which the capsomers were further aligned and classified. In the last stage, pentameric and hexameric capsomers with and without membrane were refined independently by the 'gold-standard' method. The same masks and alignment parameters were used in all of the cases. All reconstructions were low-pass filtered to 20-Å resolution for comparison.

**Fitting of atomic structures to the EM maps.** X-ray structures of Gn and Gc were flexibility fitted to the EM maps by MDFF[19]. A Gn–Gc heterodimer was built by flexible fitting of Gc (PDB:4HJ1) and rigid-body fitting of Gn into segmented density corresponding to one Gn–Gc heterodimer extracted from the localized reconstruction of the pentamer. The Gn–Gc model was symmetrized and then fitted by MDFF into localized reconstruction maps of a pentameric capsomer or hexameric capsomer types 1, 2 and 3. The same procedure was used to fit Gn and Gc (PDB:4HJC, an extended conformation) to the sub-tomogram averages of pentameric capsomer at pH 5.0 with a target membrane. A DOPC lipid bilayer model was added to the location of target membrane density. To fit models in the sub-tomogram averages of pentameric capsomer at pH 7.5 and at pH 5.0 without a target membrane, the pentameric capsomer fitted into the localized reconstruction map was used as the starting model.

MDFF runs were performed using NAMD 2.12[55] and the CHARMM36 force field[56] with the CMAP correction[57]. Simulation parameters established earlier were used[58]. AutoPSF plugin was used to prepare the structures for the simulation. This procedure created a covalent bond between all cysteine pairs, where the sulphur atoms were less than 3 Å apart. All initial configurations were solvated by TIP3P water with 0.15 M NaCl. Temperature and pressure were kept constant at 310 K and 1 bar; Langevin Piston algorithm[59] was used to keep the pressure; in the case of pentamer at pH 5.0 with target membrane, the membrane area was kept constant; particle mesh Ewald method with a grid spacing of 1 Å[60] was used to compute long-range Coulomb forces; timestep of 1 fs, non-bonded interactions evaluation step of 2 fs and full electrostatics evaluation step of 4 fs were used; water were constrained as rigid molecules; periodic boundary conditions were assumed; a scaling factor ζ = 0.3 kcal/mol; a symmetry restraint using a harmonic force constant of 200 kcal/mol/Å² in case of oligomer fitting; and secondary structure restraints were applied through all simulations to keep the cis-peptides to their current cis/trans configuration, restrain the chiral centres to their current handedness, and restrain φ, ψ dihedral angles as well as hydrogen bonds. The models were fitted into the density for 0.8–3 ns until convergence of the cross-

correlation coefficients was achieved. The geometry of all MDFF fitted models was improved by geometry minimization in PHENIX[38] and validated by Molprobity[40].

To build the atomic model for the entire icosahedral virion, an asymmetric unit composed of the 12 unique Gn–Gc heterodimers not related by icosahedral symmetry operators was fitted to the fixed virion map in UCSF Chimera (final CC of 0.96) and icosahedral symmetry was applied.

**Data availability**. Density maps, structure factors and atomic models that support the findings of this study have been deposited in the Electron Microscopy Database and in the Protein Databank with the accession codes PDB: 6F8P (RVFV Gn crystal structure), EMD-4197 and PDB: 6F9B (RVFV icosahedral reconstruction), EMD-4198 and PDB: 6F9C (RVFV hexamer 1), EMD-4199 and PDB:6F9D (RVFV hexamer 2), EMD-4200 and PDB:6F9E (RVFV hexamer 3), EMD-4201 and PDB:6F9F (RVFV pentamer). RVFV sub-tomogram averages have been deposited with the accession codes EMD-4202 and EMD-4203 (pentamer and hexamer at pH 7.5). RVFV–liposome sub-tomogram averages have been deposited with the following accession codes: EMD-4204 and EMD-4205 (pentamer and hexamer at pH 7.5 without membrane), EMD-4206 and EMD-4207 (pentamer and hexamer at pH 7.5 with membrane), EMD-4208 and EMD-4209 (pentamer and hexamer at pH 5.0 without membrane), EMD-4210 and EMD-4211 (pentamer and hexamer at pH 5.0 with membrane). All other data supporting the findings of this study are available within the article and its Supplementary Information files, or are available from the authors upon request.

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

## Acknowledgements

We thank Diamond Light Source (DLS) for access to beamline I03 (MX-14744), Mark Langridge for electron microscopy support, Jun Dong and Robert Esnouf for computing support, Robin Parsons for CL3 facility management and David Bitto for discussions. The OPIC electron microscopy facility was founded by a Wellcome Trust JIF award (060208/Z/00/Z) and is supported by a WT equipment grant (093305/Z/10/Z). This work was funded by European Research Council under the European Union's Horizon 2020 research and innovation programme (649053 to J.T.H.), Medical Research Council (MR/N002091/1 and MR/L009528/1 to T.A.B.), and a Wellcome Trust core award (090532/Z/09/Z).

## Author contributions

S.H. devised the virus purification method and together with M.L. produced the virus samples. S.H., K.H. and T.A.B. expressed, purified and solved the structure of RVFV Gn. S.H. and J.T.H. collected and processed the single-particle cryo-EM data. M.L. and S.H. performed fluorescence fusion experiments and together with S.L. prepared the virus–liposome EM grids. S.L. and S.H. collected tomography data and S.L. processed the data. S.L. advised on and together with S.H. performed MDFF. All authors contributed to writing the paper.

## Additional information

**Competing interests:** The authors declare no competing financial interests.

