## [Peer Review File · Nature Communications]

Reviewers' comments:

Reviewer #1 (Remarks to the Author):

This manuscript presents work in an important research field – virus membrane fusion, using multiple structural biology approaches including x-ray crystallography, cryo-EM, cryo-ET and subtomogram averaging techniques. The major results of this manuscript are 1) The crystal structure of the middle segment of the RVFV glycoprotein Gn shows similar topology with that in the alphavirus E2 glycoprotein. 2) The protein organization of RVFV virus was studied by the cryo-EM/3D reconstruction method to 13Å resolution. Since the virus particles are not strictly spherical based on cryo-ET reconstruction, a local capsomer averaging method was adapted to achieve better resolutions: pentamer (7.7Å), hexamers (~8-8.6Å resolutions). In addition, fitting the solved Gn protein middle component structure and the available Gc structure into the reconstruction maps lead to the conclusion that the fusion peptide in Gc is shielded by Gn. 3) cryo-ET reconstruction of virus-liposome mixture at acidic condition (pH 5.0) shows exposure of Gc to the liposomal membrane and bridge-like densities connecting between the viral and the lipid membranes. Subtomogram averaging shows that the virus interacts with the target membrane through 5-fold capsomers and the fusion peptide of the Gc protein only inserts into the outer leaflet of the target membrane.

Although much work has been done on determining part of Gn protein structure using x-ray crystallography, cryo-EM reconstruction of RVFV capsomers (result #2), the figures presented in the manuscript do not convince the readers that the reconstruction maps are at the claimed sub-nanometer resolutions and it is also difficult to examine the quality of the fitting. Further analysis and validation of the reconstruction maps and molecular modeling would improve the impact of this work in the structural biology of RVFV. I think the work on virus-liposome interaction is preliminary and need further development. My major concerns about this manuscript are:

1) The reconstruction maps do not appear to have the claimed resolutions. Several methods were employed to reconstruct the 3D maps of RVFV virus and capsomers, including single particle icosahedral reconstruction of the whole virions (13Å), localized reconstruction of pentameric and hexameric capsomers (7.7-8.6Å) and cryo-ET subtomogram averaging (14-16Å, pH7.5). However none of the figures demonstrates the quality of the reconstruction maps at nanometer or sub-nanometer resolutions. The whole virus reconstruction map (13A) was not shown. The only surface density map of the whole virus (Extended Figure 2) is a modeled map by assembling the calculated hexamers and pentamers on their projected positions. Figure 2b and 2c show very faint electron densities of the capsomers. Extended Figure 3 shows the localized reconstruction maps of the pentamer and three types of hexamer color-coded with the resolution. However these maps do not display the features corresponding to 7-8Å resolution, in which the alpha-helical structure should be obvious. Except 5-fold capsomer, the maps of hexamer do not show protein transmembrane densities. It is likely that the transmembrane domains do not appear at certain contour levels. The authors should then display cross-sections of the density maps and explain why the transmembrane protein densities are not as strong. At

~7-9Å resolution, the transmembrane domains of Gn and Gc proteins should appear as distinct cylinder-like densities at these resolutions.

2) The micrograph in Extended Data Figure 6 (E.Fig6A) shows that RVFV particles have a large size and shape variation at neutral pH. This feature likely impacts the reconstruction procedure and limits attainable resolution. I appreciate the difficulty in image processing of the virus. If the virus does not maintain strict icosahedral symmetry, or (in a worse scenario) if the number of capsomers are different for a subpopulation of the virus particles, reconstruction using icosahedral symmetry would be questionable. The authors indicated that cryo-EM experiments "required fixation to maintain an ordered icosahedral shape". There should be a more detailed description about the method of fixation and have a micrograph to demonstrate the shape and regularity of fixed particles. In addition, the extraction scheme used for localized reconstruction is also based on the assumption of icosahedral symmetry, validity of that method could also be affected if the particles have varied size and structure. The structural variability could therefore impact the quality of these localized reconstruction maps too. The authors used a subset of virus images for icosahedral reconstruction. The method section describes that 5 classes were processed for 3D classification. Do these classes show different size of particles? Is size uniformity one of the selection criteria? What are the selection criteria for the hexamers and pentamers in the localized reconstruction scheme? I think the authors should describe the nature of the particles in the revised manuscript and examine the consistency of the virus particle with icosahedral symmetry and homogeneity in the structures of different virus particles. Or if there is a previous publication discussed this issue, please cite the paper. This analysis would help to validate the results of the reconstruction maps. Tomography reconstruction and subtomogram averaging is a good approach to deal with heterogeneous particles. Consistency among the reconstruction maps from different methods help to validate the result too.

3) Because the quality of the cryo-EM reconstruction maps is not well presented, it is hard to access if the fitting is correct or not. Figures (Extended Figure 4) did not help either. Perhaps a stereoview of the fitting or a movie helps to demonstrate the quality of the fitting.

4) The manuscript described the conformation of the Gc protein in the virus structure revealed by fitting, covering of the fusion peptide by Gn, however there are no detailed discussion regards the molecular structure of the virus, ie. Gc-Gc interactions, Gn-Gn interactions. Especially the authors employ asymmetric reconstruction on hexamers, does this calculation resulted into any interesting finding about symmetry mismatch related to capsomer-capsomer interaction? The authors could extend the analysis to the protein-protein interaction on the surface of the virus and relate those findings to molecular biology studies on virus assembly and infection.

5) I think the tomography experiment on virus-liposome interactions needs more development. The tilt series has very limited tilt angles (-30 to 60). The defocus was rather high (based on E.Fig.6) and there is no description on CTF correction method. Bridge-like densities were seen in both neutral (E.Fig.6b) and low pH treated samples (E. Fig.6h)

implying these densities could be present due to non-specific binding. Fig3b shows liposomes wrapped around the virus particle suggesting a large contacting area, it is not convincing that subtomogram averaging only show close contacts at pentameric capsomers.

Other concerns:

1) Figure 2A, the size of the scale bar is missing. It is better to label two pentamers to help understanding the definition of three types of hexamers. Also it would be helpful to label pentamers and three types of hexamers in the Extended Figure 2b.

2) In the localized reconstruction method, the authors used C1, C2 and C3 symmetry for hexamers located at a general, icosahedral 2-fold and 3-fold positions. It would be helpful to describe the choice of symmetry operation for the reconstruction procedure in the method. The benefit of using these symmetry operators (in stead of simply C6) was not reflected in the manuscript.

3) Extended Figure1 uses shaded areas to indicate shared structural elements of RVFC Gn, CHIKV E2 and PUUV Gn. Using more colors might be helpful to the readers to match the similar elements among these proteins.

4) Caption for Figure 3A is not correct. This is not a slice of tomo reconstruction, in stead it is an assembled image showing modeled virus particles in a thick tomographic

Reviewer #2 (Remarks to the Author):

This is a beautiful body of work that adds much to our understanding of virus entry. The authors studied Rift Valley Fever virus, a re-energizing pathogen of consequence predicted by some to be a potential Zika in importance. The authors resent a new crystal structure of Gn which unexpectedly has a novel fold and important differences between other viruses in the family. It is most similar to alphavirus E2, suggesting an evolutionary relationship. I see that the authors used SAD. MR must have been challenged by this novel fold.

The authors then fit the crystal structure and another structure of Gc into tomographic reconstruction of visions, finding that Gc forms a kinked conformation not previously observed in class II viruses. Spectroscopic studies confirmed that a particular lipid BMP is required for fusion.

Finally, the authors show data I have waited long to see: The virus interacting with a liposome so we can see na intermediate structure in fusion. This has been a black box in viral entry for some time. The fusion loop is observed inserted into the membrane with aromatic side chains into the hydrophobic interior and polar main chains of the fusion loop interacting with the polar lipid heads. They also observe capsomere-membrane contacts to be unique to the pentamers facing the target membrane, and to not occur for the hexamers

or at neutral-pH.

The combined structural biology approaches suggest a shielding mechanism of the Gc fusion loops by the Gn partner, and propose important models for other class II viruses for which such insight is lacking.

This is a highly significant body of work that informs multiple threats to human health.

I would have expected to review such work for Nature and feel that it is worthy of such high-tier journals. It is an excellent contribution to the field. It is exciting and timely.

Reviewer #3 (Remarks to the Author):

In this manuscript by Halldorsson et. al. the authors describe an integrated structural study on the envelope glycoprotein's (Gn and Gc) organization and conformational changes during the early stages of Rift Valley Fever Virus (RVFV) infection. The authors determine the crystal structure of Gn at atomic resolution and by combining available structures of Gc, Cryo-EM reconstruction and molecular dynamic they proposed a model for the assembly of the virus particle as well as suggested mechanism for the structural changes in the envelope preceding and during the membrane fusion step. The manuscript is well written and presents with high quality illustrations. The mode of RVFV assembly and the structural changes during early events of infection will be beneficial to the field as well as related disciplines. However, although this work presents an alternative model from RVFV glycoprotein's structure and virion assembly and provides new aspects into the events preceding and the early steps of membrane fusion there are some major concerns regarding this work:

Major Points –

1. Concerning the crystallographic work presented in this manuscript there is no novelty as the structures of RVFV Gn and the related SFTSV Gn were published recently in PNAS (Wu et.al 2017). The relation to Alphavirus' E2 fold was also demonstrated in that publication. Therefore, the authors should at least cite this paper and/or discuss any differences between their structure and the published one.
2. The authors claim to structural plasticity in domain I of Gc as opposed to the previously published structures RVFV Gc. Although a hinge motion was observed in the E1 structure (Ref. 17) it is more of a moderate rigid body movement rather than tertiary structure deformation presented here. Indeed, the authors provide a view on the fitted model in their cryo-EM volume however this fitting seems unpersuasive at the molecular level (for example; glycan positioning was ignored or disulfide bonds that rigidify the structure were not addressed). This imply that this proposed novel conformation of Gc will benefit from experimental validation either by higher resolution data (that at least secondary structure could be observed) or by other biochemical methods such as immunogold labeling and/or mutagenesis.
3. The low pH cryo-tomograms that implying the mode of membrane penetration of Gc is

unconvincing at that resolution. There is significantly large unoccupied volume at the viral-membrane proximal end of the pentamer (does that imply for Gc domain III or stem region flexibility?) and from the density of the liposome membrane proximal end of the capsomer it seems that Gc might be oriented alternatively.

Minor points –

1. It would be helpful for the readers to have a comparative molecular view on Gn-Gc heterodimers both in pentamer versus hexamer as well as neutral versus low pH.
2. In a work by Rönkä et.al. (1995) on Uukuniemi virus it was shown that Gn-Gc complexes are pH depended and they fall apart starting at pH 6.2. In their low pH capsomer model, does the authors observe an interaction interface between Gn and Gc? If so, what are the steric implication for such interactions to the conversion of Gc into the post-fusion trimer?
3. The authors should discuss alternative explanation to the intense dimerization interface in the anti-parallel dimer structure and the homotypic manner of the fusion loop shielding that was observed in the high resolution structure reported in Ref. 3
4. It seems that in all their molecular graphics the authors did not refer to the glycan chains of Gc. Do the authors have hypothesis for the position and role of Gc's glycans in their assembly model?
5. It was previously reported of Gc oligomerization in pH as low as 5.5 (Ref. 23) and also other class II membrane fusion protein were reported to adopt the parallel trimer conformation after low pH treatment in the presence of liposomes. Do the authors observe evidence to such structures? And if not, do they have hypothesis why not?

Given all of the above I would recommend that this manuscript will be accepted for publication in Nature Communications upon major revisions

We would like to thank all of the Reviewers for their accurate summary of our results and constructive criticism. We were very pleased that the Reviewers thought that this was “an important research field” (Reviewer #1), “highly significant body of work” (Reviewer #2) and “well written... with high quality illustrations” (Reviewer #3). As requested, we have performed additional analyses to support the findings of this study and provide detailed responses for each of their points below. Please note that our text is coloured blue and the Reviewers' comments are in black.

Reviewers' comments:

Reviewer #1 (Remarks to the Author):

This manuscript presents work in an important research field – virus membrane fusion, using multiple structural biology approaches including x-ray crystallography, cryo-EM, cryo-ET and subtomogram averaging techniques. The major results of this manuscript are 1) The crystal structure of the middle segment of the RVFV glycoprotein Gn shows similar topology with that in the alphavirus E2 glycoprotein. 2) The protein organization of RVFV virus was studied by the cryo-EM/3D reconstruction method to 13Å resolution. Since the virus particles are not strictly spherical based on cryo-ET reconstruction, a local capsomer averaging method was adapted to achieve better resolutions: pentamer (7.7Å), hexamers (~8-8.6Å resolutions). In addition, fitting the solved Gn protein middle component structure and the available Gc structure into the reconstruction maps lead to the conclusion that the fusion peptide in Gc is shielded by Gn. 3) cryo-ET reconstruction of virus-liposome mixture at acidic condition (pH 5.0) shows exposure of Gc to the liposomal membrane and bridge-like densities connecting between the viral and the lipid membranes. Subtomogram averaging shows that the virus interacts with the target membrane through 5-fold capsomers and the fusion peptide of the Gc protein only inserts into the outer leaflet of the target membrane.

Although much work has been done on determining part of Gn protein structure using x-ray crystallography, cryo-EM reconstruction of RVFV capsomers (result #2), the figures presented in the manuscript do not convince the readers that the reconstruction maps are at the claimed sub-nanometer resolutions and it is also difficult to examine the quality of the fitting. Further analysis and validation of the reconstruction maps and molecular modeling would improve the impact of this work in the structural biology of RVFV. I think the work on virus-liposome interaction is preliminary and need further development.

My major concerns about this manuscript are:

1) The reconstruction maps do not appear to have the claimed resolutions. Several methods were employed to reconstruct the 3D maps of RVFV virus and capsomers, including single particle icosahedral reconstruction of the whole virions (13Å), localized reconstruction of pentameric and hexameric capsomers (7.7-8.6Å) and cryo-ET subtomogram averaging (14-16Å, pH7.5). However none

of the figures demonstrates the quality of the reconstruction maps at nanometer or sub-nanometer resolutions.

We apologise to the Reviewer for having omitted the validation analysis of the reported resolution values in the manuscript. The FSC (Fourier shell correlation) curves that we are submitting as part of the map submission to EMDB have now also been included in the manuscript for each of the 15 reconstructions reported in this manuscript (Supplementary Figures 2, 4 and 10). These curves demonstrate how the claimed resolution for each of reconstruction maps was estimated using the 0.143-threshold after 'gold-standard' refinement of each of the structures.

We would like to note that the sub-tomogram averages in Supplementary Figure 9 had all been low-pass filtered to 20-Å resolution to allow their direct comparison. We apologise for having omitted this detail, which has now been added to the figure legend: "All sub-tomogram averages have been low-pass filtered to 20 Å resolution for comparison."

Unfiltered versions of these sub-tomogram averages are shown together with the respective FSC curves in Supplementary Figure 10.

Finally, in-line to our response to Reviewer 3, we have also added a movie (Supplementary movie 1), which in addition to Supplementary Figure 6 illustrates the secondary structure features expected to be resolved at the resolution of the localized reconstructions (i.e. separation of β -sheets at resolution of 7.7–8.6 Å). We would like to note that there are very few α -helices present but these are resolved as can be seen in the movie. The couple of very short α -helices that are present are resolved as can be seen in the movie.

In conclusion, our structural features are fully consistent with the resolutions reported and indicated by the FSC plots.

The whole virus reconstruction map (13A) was not shown.

We apologise for rendering the whole virion reconstruction density as a transparent surface, making it difficult to visualize (originally Fig. 2a). We have taken this opportunity to improve our figure presentation. In particular, we now show the reconstruction of the full virion in opaque rendering (new Fig. 2a,b; Supplementary Figure 2).

The only surface density map of the whole virus (Extended Figure 2) is a modeled map by assembling the calculated hexamers and pentamers on their projected positions. Figure 2b and 2c show very faint electron densities of the capsomers.

As described above, we have now added the surface density map of the whole virus (new Fig. 2a,b; Supplementary Figure 2).

Extended Figure 3 shows the localized reconstruction maps of the pentamer and three types of hexamer color-coded with the resolution. However these maps do not display the features corresponding to 7-8Å resolution, in which the alpha-helical structure should be obvious.

As noted above, the ectodomains of RVFV Gn and Gc glycoproteins present very limited α -helical content and the transmembrane α -helices were not resolved as well as the rest of the structure (please see below). However, the separation between β -sheets was clearly visible, in line with the expected resolution. As described above, we have included the FSC curves for these reconstructions, which also confirm the estimated resolutions (Supplementary Figure 4).

Except 5-fold capsomer, the maps of hexamer do not show protein transmembrane densities. It is likely that the transmembrane domains do not appear at certain contour levels. The authors should then display cross-sections of the density maps ...

We thank the Reviewer for making this very relevant point. As requested, we have added an additional supplementary figure that illustrates cross-sections of the density maps (Supplementary Figure 3).

... and explain why the transmembrane protein densities are not as strong. At $\sim 7-9\text{\AA}$ resolution, the transmembrane domains of Gn and Gc proteins should appear as distinct cylinder-like densities at these resolutions.

We would like to note that the resolution achieved in the membrane portion of our reconstructions is lower than 9 Å, as illustrated by our local resolution analysis (Supplementary figure 4). We suggest that this lower resolution may be attributed to the higher degree of disorder in the membrane regions. Due to this disorder, the transmembrane densities do not appear in the isosurface representation, although most of them are faintly visible in the cross-sections (Supplementary Figure 3).

2) The micrograph in Extended Data Figure 6 (E.Fig6A) shows that RVFV particles have a large size and shape variation at neutral pH. This feature likely impacts the reconstruction procedure and limits attainable resolution. I appreciate the difficulty in image processing of the virus.

We agree with the Reviewer that the inherent flexibility of RVFV limited the resolution that could be achieved by adopting a conventional icosahedral reconstruction strategy. Hence, we resorted to Localized Reconstruction, a methodology we have previously developed to tackle this type of problem.

If the virus does not maintain strict icosahedral symmetry, or (in a worse scenario) if the number of capsomers are different for a subpopulation of the virus particles, reconstruction using icosahedral symmetry would be questionable.

We thank the Reviewer for the opportunity to clarify this point. We have now added additional data, which verifies that the number of capsomers is consistent with one type of icosahedral symmetry (Supplementary Figure 2). This includes class averages (both 2D and 3D).

The authors indicated that cryo-EM experiments “required fixation to maintain an ordered icosahedral shape”. There should be a more detailed description about the method of fixation

As requested, we have added more detail on the fixation methods. The following text has been added in the Results:

Page 3, lines 30–33: “In our attempts to preserve the shape and icosahedral symmetry during purification, we devised an improved sample preparation strategy based on fixation of the particles with formaldehyde by dialysis, followed by sucrose gradient purification (see Methods; Supplementary Fig 2c,d).”

The fixation method is explained in detail in the Methods section:

Page 7, lines 26–32: “To better preserve the virion ultrastructure for single particle cryo-EM, the supernatant was collected from the infected cells as above. The virus particles were chemically fixed by dialysing the supernatant against PBS containing 0.2% v/v formaldehyde for ~20 hours. Excess formaldehyde was removed by dialysing against PBS. The fixed virus was concentrated by reverse dialysis using solid 35 kDa PEG and purified by gradient ultracentrifugation on a 20–60 % w/v sucrose gradient in T20N100 buffer. Gradient fractions were harvested using a gradient fractionator.”

and have a micrograph to demonstrate the shape and regularity of fixed particles.

As requested, we have now added a figure showing that some, but not all, of the fixed RVFV particles retain their icosahedral structure upon purification (Supplementary Figure 2c). 2D class-averages of the most-ordered particles selected for image processing show views consistent with icosahedral symmetry (Supplementary Figure 2e).

In addition, the extraction scheme used for localized reconstruction is also based on the assumption of icosahedral symmetry, validity of that method could also be affected if the particles have varied size and structure. The structural variability could therefore impact the quality of these localized reconstruction maps too. The authors used a subset of virus images for icosahedral reconstruction. The method section describes that 5 classes were processed for 3D classification. Do these classes show different size of particles?

We completely agree with the Reviewer that different sizes of particles would affect both conventional single particle reconstructions and the subsequent localized reconstructions of the individual capsomers. However, this was not the

case: Our 3D classification showed that all particles were consistent with the assembly of T=12 icosahedrons. To illustrate this point, we have added 2D and 3D class averages to Supplementary Figure 2 and added the following sentence to the Methods section to describe how we selected particles for the 13-Å reconstruction of the full virion:

Page 9, lines 19–24: “To extract an ordered subset of particles for final refinement, particles were binned by a factor of two (box size of 512×512 pixels with pixel size of 2.7 Å) and subjected to 3D classification to 10 classes. No further alignment was performed in this final round of classification to speed up the computations. A subset of 2,995 particles was selected from the three most ordered classes (Supplementary Fig 2e)”.

Is size uniformity one of the selection criteria?

As we have noted above, there was no major size variation in the RVFV particles selected for 3D refinement. The selection criteria used was the consistent appearance of the 3D classes. The classes chosen are shown now in Supplementary Figure 2 and this is mentioned in the figure legend.

What are the selection criteria for the hexamers and pentamers in the localized reconstruction scheme?

The selection criteria used was the resolution of the 3D classes of pentamers and hexamers. We have added this information to the Methods section:

Page 9, lines 41–43: “Sub-particles from the highest resolution 3D classes were selected for further 3D refinement.”

I think the authors should describe the nature of the particles in the revised manuscript and examine the consistency of the virus particle with icosahedral symmetry and homogeneity in the structures of different virus particles.

We thank the Reviewer for raising these important points. As we have explained in the comments above and presented in our added supplementary data (Supplementary Figure 2), our additional analyses demonstrate that despite the observed flexibility of RVFV particles, a subset of them sufficiently retain icosahedral symmetry and are amenable to icosahedral averaging.

Or if there is a previous publication discussed this issue, please cite the paper.

The suitability of RVFV particles has also been shown in earlier papers that used single particle icosahedral reconstruction (Huiskonen et al 2009, J Virol; Sherman et al 2009 Virology), albeit at much lower resolution. Both of these papers are cited in the manuscript.

This analysis would help to validate the results of the reconstruction maps.

As we have iterated in our response, we now provide several new types of validation analysis, including the comparison between three different reconstruction methods suggested by the Reviewer (see next point). Also the consistency between the 12 structures making the pentamers and hexamers in the T=12 icosahedral lattice provides an additional internal control in the icosahedral single particle reconstruction, as these densities are independent from one another. This is now visible in Supplementary Movie 1 and new Figure 2a,b.

Tomography reconstruction and subtomogram averaging is a good approach to deal with heterogeneous particles. Consistency among the reconstruction maps from different methods help to validate the result too.

We completely agree with the Reviewer and are thankful for this valuable suggestion. We have now calculated additional subtomogram averages of the pentamer and hexamer at neutral pH at 14-Å resolution and have compared these structures to the single particle whole virion reconstruction and the localized reconstruction (Supplementary Figure 5). Consistency between these reconstructions further validates our results.

3) Because the quality of the cryo-EM reconstruction maps is not well presented, it is hard to access if the fitting is correct or not. Figures (Extended Figure 4) did not help either. Perhaps a stereoview of the fitting or a movie helps to demonstrate the quality of the fitting.

We thank the Reviewer for this constructive criticism. We also agree that a more three-dimensional representation would be beneficial to illustrate this large and quite complex structure. As also requested by Reviewer 3, we added a movie to better show the fitting and to further demonstrate the quality of the reconstructed maps (Supplementary Movie 1). We hope that it is clear now that the secondary structure elements including β -sheets and the few α -helices that are present in the structure all fit well into the density.

4) The manuscript described the conformation of the Gc protein in the virus structure revealed by fitting, covering of the fusion peptide by Gn, however there are no detailed discussion regards the molecular structure of the virus, ie. Gc-Gc interactions, Gn-Gn interactions

This is a very interesting point and we completely agree that an understanding of homo- and heterotypic RVFV glycoprotein interactions is essential for understanding virion assembly. We would like to note, however, that whilst the resolution of our analysis was sufficient to reveal the covering of the fusion peptide by Gn, it would be too speculative to describe more detailed interactions, as suggested by the Reviewer. Also, the membrane-proximal parts of the Gn have not been structurally characterised by crystallography and these may play a role in some of the capsomer-capsomer interactions.

Especially the authors employ asymmetric reconstruction on hexamers, does this calculation resulted into any interesting finding about symmetry mismatch

related to capsomer-capsomer interaction? The authors could extend the analysis to the protein-protein interaction on the surface of the virus and relate those findings to molecular biology studies on virus assembly and infection.

In this work, we focused on early steps in membrane fusion and not virus assembly. In this response we present data that highlights the quasi-equivalence of the Gn-Gc heterodimers (see Figure 1 in our response to Reviewer 3). Improving the resolution of the capsomers to facilitate more detailed analysis of these interactions is an exciting area of research and is something we intend to pursue in our future work.

5) I think the tomography experiment on virus-liposome interactions needs more development. The tilt series has very limited tilt angles (-30 to 60).

We agree that the tilt range used is indeed an important consideration in electron cryo-tomography; the narrower the tilt range, the larger the missing wedge. However, we would like to note that in our subtomogram averaging approach, which combines subtomograms in fully random orientations around the virions, the missing wedge is averaged out completely, resulting in uniform resolution of the averages.

We would like to suggest that in our case, the moderately limited tilt range is actually beneficial, as increasing the range would increase the electron dose and also the induced radiation damage. Indeed, these considerations are consistent for example with the following publication, where similar or even more limited tilt range (from -45° to +45° or from -30° to +30°) was combined with subtomogram averaging to achieve sub-nanometer resolution:

Schur FKM, Dick RA, Hagen WJH, Vogt VM, Briggs JAG. 2015. The Structure of Immature Virus-Like Rous Sarcoma Virus Gag Particles Reveals a Structural Role for the p10 Domain in Assembly. *J Virol* 89:10294–10302.

The defocus was rather high (based on E.Fig.6)

As no phase-plate was available, a range of relatively large defocus values was required to increase the contrast and facilitate alignment of the capsomers. However, the values we used are not out of the ordinary. The maximum defocus we had chosen (4.0 μm ; Supplementary Table 3) is actually smaller than that used by Schur et al. (4.5 μm) who, despite the fact, achieved sub-nanometer resolution.

and there is no description on CTF correction method.

We apologise for this omission and have now added the following details of the CTF correction applied and clarified also how CTF was estimated in the reconstruction to the Methods section:

Page 9, lines 47–48 and page 10, lines 1–3: “Tilt series were dose-weighted by taking into account the accumulated dose (ref 50). Unweighted, raw tilt series

were used to estimate the defocus parameters using Gctf (ref 51). The estimated defocus was used to correct the CTF of the dose-weighted tilt series by *ctfphaseflip* in IMOD using strip width of 20 pixels (ref 52).”

Bridge-like densities were seen in both neutral (E.Fig.6b) and low pH treated samples (E. Fig.6h) implying these densities could be present due to non-specific binding.

We thank the Reviewer for the opportunity to clarify our description of the RVFV–liposome interactions. As shown in Supplementary Figure 8, we observed extensive virus–liposome contact areas at both neutral and acidic pH. As the Reviewer correctly observes, the pH-neutral control and acidified samples appear similar in both cryo-EM images and cryo-ET slices. It is only upon subtomogram averaging where differences in the spike–liposome membrane interactions are revealed (please see our next point below). To clarify this point, we have modified the text in the figure legend as follows:

“Whilst the contact zones occur both at pH-neutral control and in the acidified samples, subtomogram averages reveal the membrane insertion of the pentameric capsomers only in the acidified sample (j).”

Fig3b shows liposomes wrapped around the virus particle suggesting a large contacting area, it is not convincing that subtomogram averaging only show close contacts at pentameric capsomers.

As discussed above, we agree that it is difficult to discern differences in structure from electron tomographic models alone. It is only by subtomogram averaging that we were able to differentiate the pentamer–liposome and hexamer–liposome contacts. As described in the main text, and as illustrated in the Supplementary Figure 9, it is indeed only the averaged pentamer that forms ordered contacts with the liposome membranes.

However, as the Reviewer correctly suggests, it is possible that a sub-population of the hexamers have undergone similar conformational changes (and conversely, that some of the pentamers have not undergone these conformational changes). Such changes would not be observable by our averaging method if the fraction of spikes in an alternative conformation(s) was low. We have added the following to Discussion section to reflect this caveat:

Page 6, lines 13–21: “Interestingly, the embedding of the fusion loops was only observed in the case of the pentamers and not with the hexamers. While it is tempting to speculate that the pentamers might be energetically in a more metastable state than the hexamers, possibly due to less favorable subunit–subunit angles (72° in the 60 heterodimers making the pentamers as opposed to 60° in the 660 heterodimers making the hexamers) this hypothesis remains to be tested. Furthermore, by our sub-tomogram averaging approach we cannot exclude the possibility that a small population of the hexamers is also engaged with the target membrane, or conversely, that a small population of the pentamers facing the membrane is still in a prefusion state.”

Other concerns:

1) Figure 2A, the size of the scale bar is missing.

We have added the lengths of the scale bars in the figure legend.

It is better to label two pentamers to help understanding the definition of three types of hexamers.

Modified as suggested.

Also it would be helpful to label pentamers and three types of hexamers in the Extended Figure 2b

Modified as suggested.

2) In the localized reconstruction method, the authors used C1, C2 and C3 symmetry for hexamers located at a general, icosahedral 2-fold and 3-fold positions. It would be helpful to describe the choice of symmetry operation for the reconstruction procedure in the method. The benefit of using these symmetry operators (in stead of simply C6) was not reflected in the manuscript.

We thank the Reviewer for this point. For the localized reconstruction method, we tested C1, C2, C3, and C6 symmetries for each of the hexamers. Interestingly, it was the original local symmetry (or no symmetry) of the icosahedral reconstruction that always gave the highest resolution also for the localized reconstruction. The lack of symmetry of type 1 hexamer (general location), prominent two-fold symmetry of the type 2 hexamer (2-fold location) and prominent three-fold symmetry of type 3 hexamer (3-fold location) can now be seen in new Supplementary Figure 3. These symmetry operators were then used to calculate the final maps.

We have now added this in the Results section: "Imposing lower than six-fold symmetry (two-fold symmetry for the type 2 hexamer and three-fold symmetry for the type 3 hexamer), or no symmetry (type 1 hexamer) gave the highest resolution, suggesting significant deviations from six-fold symmetry in the hexamers (Supplementary Figure 3)."

3) Extended Figure1 uses shaded areas to indicate shared structural elements of RVFC Gn, CHIKV E2 and PUUV Gn. Using more colors might be helpful to the readers to match the similar elements among these proteins.

We were interested to note that RVFC Gn, CHIKV E2 and PUUV Gn exhibit similar patterns of secondary structure. However, although we would be keen to update the figure as suggested, we feel that this would lead to an over-interpretation due to the high level of tertiary structure deviation.

4) Caption for Figure 3A is not correct. This is not a slice of tomo reconstruction,

in stead it is an assembled image showing modeled virus particles in a thick tomographic

We thank the Reviewer for spotting this textual error. The figure legend has been corrected.

Reviewer #2 (Remarks to the Author):

This is a beautiful body of work that adds much to our understanding of virus entry. The authors studied Rift Valley Fever virus, a re-energizing pathogen of consequence predicted by some to be a potential Zika in importance. The authors resent a new crystal structure of Gn which unexpectedly has a novel fold and important differences between other viruses in the family. It is most similar to alphavirus E2, suggesting an evolutionary relationship. I see that the authors used SAD. MR must have been challenged by this novel fold.

The authors then fit the crystal structure and another structure of Gc into tomographic reconstruction of visions, finding that Gc forms a kinked conformation not previously observed in class II viruses. Spectroscopic studies confirmed that a particular lipid BMP is required for fusion.

Finally, the authors show data I have waited long to see: The virus interacting with a liposome so we can see na intermediate structure in fusion. This has been a black box in viral entry for some time. The fusion loop is observed inserted into the membrane with aromatic side chains into the hydrophobic interior and polar main chains of the fusion loop interacting with the polar lipid heads. They also observe capsomere-membrane contacts to be unique to the pentamers facing the target membrane, and to not occur for the hexamers or at neutral-pH.

The combined structural biology approaches suggest a shielding mechanism of the Gc fusion loops by the Gn partner, and propose important models for other class II viruses for which such insight is lacking.

This is a highly significant body of work that informs multiple threats to human health.

I would have expected to review such work for Nature and feel that it is worthy of such high-tier journals. It is an excellent contribution to the field. It is exciting and timely.

We are delighted that the Reviewer shares our enthusiasm for this work and are thankful for the highly encouraging comments.

Reviewer #3 (Remarks to the Author):

In this manuscript by Halldorsson et. al. the authors describe an integrated structural study on the envelope glycoprotein's (Gn and Gc) organization and conformational changes during the early stages of Rift Valley Fever Virus (RVFV) infection. The authors determine the crystal structure of Gn at atomic resolution

and by combining available structures of Gc, Cryo-EM reconstruction and molecular dynamic they proposed a model for the assembly of the virus particle as well as suggested mechanism for the structural changes in the envelope preceding and during the membrane fusion step. The manuscript is well written and presents with high quality illustrations. The mode of RVFV assembly and the structural changes during early events of infection will be beneficial to the field as well as related disciplines.

However, although this work presents an alternative model from RVFV glycoprotein's structure and virion assembly and provides new aspects into the events preceding and the early steps of membrane fusion there are some major concerns regarding this work:

Major Points –

1. Concerning the crystallographic work presented in this manuscript there is no novelty as the structures of RVFV Gn and the related SFV Gn were published recently in PNAS (Wu et.al 2017). The relation to Alphavirus' E2 fold was also demonstrated in that publication. Therefore, the authors should at least cite this paper and/or discuss any differences between their structure and the published one.

We thank the Reviewer for this point and have added a reference to the other RVFV Gn structure, which was very recently reported. As requested, we have also added discussion comparing these similar structures, and how this similarity is suggestive of a conserved mode of assembly for phleboviral glycoproteins more broadly:

Page 5, lines 38–45: “Recently Wu *et al.* reported the crystal structure RVFV Gn glycoprotein ectodomain N-terminal fragment (ref 31). Superposition analysis revealed a high level of structural similarity between that Gn structure and the one reported in this study (0.4 Å root-mean square deviation between 261 pairs of C-alpha atoms, respectively). In light of the preserved nature of this fold architecture and another phlebovirus, severe fever with thrombocytopenia syndrome virus (ref 31), we propose that our model of RVFV Gn–Gc heterodimerisation and higher order assembly (Fig. 2c–e) is likely to be observed across the viruses within the *phlebovirus* genus. “

In line with this modification, we have also added a comment describing how our higher resolution EM map, reported here, enabled a more confident localization of the Gn and Gc glycoproteins within the virion envelope, than previously published:

Page 5, lines 45–46 and page 6, lines 1–4 “Furthermore, our work also clarifies earlier hypotheses on the placement of the Gn (ref 31) and Gc (ref 11) in the RVFV virion. Indeed, the improved resolution of the density maps for the capsomers reported here (7.7–8.6 Å), with respect to our earlier 22-Å resolution reconstruction (ref 2), allowed the construction of an accurate atomic model of the whole virion surface (Fig 2, Supplementary Movie 1).”

2. The authors claim to structural plasticity in domain I of Gc as opposed to the previously published structures RVFV Gc. Although a hinge motion was observed in the E1 structure (Ref. 17) it is more of a moderate rigid body movement rather than tertiary structure deformation presented here.

We agree with the Reviewer that the structural plasticity of the Gc, observed in our EM map, is surprising and the scale of this motion is larger than in the E1 structure. In the light of these considerations, we have revised our statement as follows:

Page 4, lines 37–41: “Whilst such degree of conformational plasticity is unprecedented in the known class II fusion proteins, we would like to note that to a lesser extent plasticity in the same region has been observed to be necessary for the icosahedral assembly of the fusion glycoprotein in an alphavirus (ref 24)”

Indeed, the authors provide a view on the fitted model in their cryo-EM volume however this fitting seems unpersuasive at the molecular level (for example; glycan positioning was ignored...

As suggested, we have performed additional analyses of the glycan positioning to further confirm the Gn and Gc fitting. We have added a new figure (Supplementary Figure 7) and the following text in the Results section:

Page 4, lines 26–33: “As the N-linked glycosylation displayed on viruses is unlikely to be obscured in protein–protein interfaces (refs 22,23), mapping of putative N-linked glycosylation sites onto pentameric and hexameric Gn–Gc capsomers was performed to validate the model. This analysis revealed all putative N-linked glycosylation sites to be either solvent exposed (RVFV Gn Asn438 and RVFV Gc Asn794, Asn1035) or facing unoccupied density that could partially correspond to an ordered glycan (Gc site Asn1077; Supplementary Figure 7).

...or disulfide bonds that rigidify the structure were not addressed).

We thank the reviewer for the opportunity to clarify the technical point regarding the disulphide bonds. We would like to note that these bonds were not ignored. Instead, they were explicitly defined as covalent bonds in the molecular dynamics flexible fitting, and they were retained as such during the course of the simulation. This important detail has been added in the Methods:

Page 10, lines 47–82 and page 11, line 1: “AutoPSF plugin was used to prepare the structures for the simulation. This procedure created a covalent bond between all cysteine pairs where the sulphur atoms were less than 3 Å apart. “

This imply that this proposed novel conformation of Gc will benefit from experimental validation either by higher resolution data (that at least secondary structure could be observed)...

We thank the Reviewer for the opportunity to further clarify this point. As we have explained in our responses to Reviewer 1, our localized reconstruction of the pentamer reached resolution of 7.7 Å, which was sufficient to observe most of the secondary structure elements present in the structure, notably clear separation of the β-sheets of the Gc glycoprotein. This allowed unambiguous fitting of the Gc glycoprotein. As we have noted above, there is very little α-helical content in these molecules.

We agree that higher resolution data would indeed clarify the detailed molecular interactions between the RVFV Gn and Gc glycoproteins but due to the flexibility of the RVFV virions this has not yet been possible. Extending the resolution with improved localized reconstruction methods and/or sub-tomogram averaging is something we will pursue in our further studies.

...or by other biochemical methods such as immunogold labeling and/or mutagenesis.

While the biochemical experiments suggested by the Reviewer are beyond the scope of this study, we have added a movie (Supplementary Movie 1) to illustrate the unambiguous fit in more detail (this was also requested by Reviewer 1).

3. The low pH cryo-tomograms that implying the mode of membrane penetration of Gc is unconvincing at that resolution.

We agree that higher resolution would increase the accuracy of fitting. However, to date this has not been possible and, to the best of our knowledge, our results constitute the first membrane penetration fusion intermediate resolved. We have added discussion on the limitations of the fitting due to limited resolution in the Discussion (see below).

There is significantly large unoccupied volume at the viral-membrane proximal end of the pentamer (does that imply for Gc domain III or stem region flexibility?)...

We thank the Reviewer for bringing this point to our attention and would like to note that substantial parts of the Gn and Gc membrane-proximal regions are absent from the reported crystal structures, as these structures were derived from truncated ectodomains. We thus suggest that the unoccupied density corresponds collectively to these regions, whose structures are unknown. We have updated Figure 4c to reflect this and have modified the figure legend as follows:

“The locations of the two leaflets in both the viral and the liposomal membrane are indicated with a dashed line. Unoccupied densities, assigned to C-terminal parts of the Gc and Gn glycoproteins that are absent from the fitted crystal structures, are indicated with asterisks.”

...and from the density of the liposome membrane proximal end of the capsomer it seems that Gc might be oriented alternatively.

We completely agree with the Reviewer that crystal structure fitting into maps at this resolution (20 Å) may result in some ambiguity, depending on the shape of the protein. Here, the extended form of the Gc glycoprotein is rod-shaped and thus we cannot preclude the existence of alternative orientations, where it may be somewhat rotated around its long axis. However, the dimensions of the molecule, combined with the membrane-to-membrane distance unequivocally confirm that the density proximal to the liposome membrane corresponds to the fusion loops. In line with this comment, we now write in the Discussion:

Page 6, lines 5–11: “Our model of the early RVFV fusion intermediate unequivocally shows that the fusion loops of the Gc glycoprotein embed in the host membrane at low pH. The accuracy of the fitted model, however, is limited by the relatively low resolution of the sub-tomogram average of the membrane facing pentamer (20 Å). In general, rigid-body fitting to maps to comparable resolutions can reach an accuracy as high as ~4 Å (ref 32). In our case, the accuracy is expected to be lower, due to the fact that flexible fitting was required.”

Minor points –

1. It would be helpful for the readers to have a comparative molecular view on Gn-Gc heterodimers both in pentamer versus hexamer...

We agree that this is an interesting point. Quasi-equivalence of the Gn-Gc heterodimers is evident in the structure (see attached Figure 1). This phenomenon is related to the assembly of the flexible RVFV virions, a topic we are currently pursuing in parallel to our current study.

Figure 1. Quasi-equivalence of the Gn-Gc heterodimers.

...as well as neutral versus low pH.

As requested, we have added this comparison as Supplementary Figure 11. Given that some ambiguity still exists in the orientation of Gn at low pH, we have added the caveat to the Discussion to avoid over-interpretation of the results:

Page 6, lines 11–12: "...the molecular details of the major conformational change in the Gn–Gc complex remain to be addressed in further studies at higher resolution."

2. In a work by Rönkä et.al. (1995) on Uukuniemi virus it was shown that Gn-Gc complexes are pH depended and they fall apart starting at pH 6.2. In their low pH capsomer model, does the authors observe an interaction interface between Gn and Gc?

We thank for the Reviewer for noting this early reference. We would like clarify that this publication solely reports the detection of homomers of Uukuniemi virus glycoproteins (formerly referred to as G1 and G2), suggesting that either the techniques used were unable to resolve the heterodimeric assembly or that Uukuniemi virus assembles in a distinct manner. As we have noted above, we would like to refrain from inferring the specific molecular details of the Gn–Gc interaction at low-pH given the limited resolution of the reconstruction.

If so, what are the steric implication for such interactions to the conversion of Gc into the post-fusion trimer?

This is very interesting point. Although it is not addressable from our current data, which characterizes an early fusion intermediate, this is something we are very keen to pursue as a part of our future work.

3. The authors should discuss alternative explanation to the intense dimerization interface in the anti-parallel dimer structure and the homotypic manner of the fusion loop shielding that was observed in the high resolution structure reported in Ref. 3

It is indeed interesting that many class II fusion protein ectodomains assemble as homodimers in solution or in a crystalline state. However, we note that such a homodimeric state is not always a physiologically relevant. For example, alphaviruses, such as chikungunya virus (Voss et al. 2010 Nature) and Semliki Forest virus (Lescar et al. 2001 Cell), form heterotypic organizations, such as those observed in our current work for RVFV.

We would like to highlight that we have emphasized the similarities between the alphaviruses mentioned above and RVFV in the Results section as follows:

Page 4, lines 42–47: "The Gn–Gc interface is formed between the β -ribbon domain of the Gn hydrophobic fusion loops of Gc (ref 11) are shielded from solvent at the interface between Gn domains A and B (Fig. 2f). This is reminiscent to alphaviruses SFV and CHIKV, where the fusion loop of the class II fusion

protein E1 protein is shielded by the E2 partner protein (Fig. 2f) (refs 17,24,25) but contrasts the homotypic shielding observed in flaviviruses such as ZIKV (ref 26) and DENV (refs 10,27).“

4. It seems that in all their molecular graphics the authors did not refer to the glycan chains of Gc.

As described above, we have now analysed the position of N-linked glycosylation sequons to validate and support our fitting (Supplementary Figure 7).

Do the authors have hypothesis for the position and role of Gc's glycans in their assembly model?

The Reviewer is correct that N-linked glycans are likely to play a role in the functionality of the Gn-Gc spike complex, particularly in receptor binding but possibly also in assembly. We note, however, that the composition and occupancy of the N-linked glycosylation on both these glycoproteins has yet to be established. In the absence of any data, it would be premature to hypothesize the specific role of individual glycan sites.

5. It was previously reported of Gc oligomerization in pH as low as 5.5 (Ref. 23) and also other class II membrane fusion protein were reported to adopt the parallel trimer conformation after low pH treatment in the presence of liposomes. Do the authors observe evidence to such structures? And if not, do they have hypothesis why not?

Based on existing models of class II fusion rearrangements (Harrison 2008. NSMB; Modis 2014. Current Opin Virol), the Gc glycoprotein is expected to trimerize following full fusion of the viral and host cell membranes. In our manuscript, we have solely focused on the early steps of this process and incubated our virus-liposome mixtures for only ~30 s. After this incubation, we did not observe full fusion, and consistently, we did not observe the formation of trimeric Gc structures in our data. To be able to observe such post-fusion trimers in our system, the fusion process would be required to proceed until the very end, where the viral and liposomal membranes are fully fused. We hypothesize that this takes longer than the 30-s incubation used to trap the early intermediate presented in this study. Later time points in the fusion pathway remain under our active investigation. We now write in the Discussion: Page 6, lines 29-33: “How Gc proteins merge the target membrane into the viral membrane, the number of Gc monomers that are involved in merging the membranes, and how they assume a trimeric conformation in the fused membrane (refs 1,10,12) remains to be addressed by structural analysis of later intermediates.”

Given all of the above I would recommend that this manuscript will be accepted for publication in Nature Communications upon major revisions

Once again, we would like to iterate our thanks for the thoughtful comments of each Reviewer.

REVIEWERS' COMMENTS:

Reviewer #3 (Remarks to the Author):

In this manuscript by Halldorsson et. al. the authors describe an integrated structural study on the envelope glycoprotein's (Gn and Gc) organization and conformational changes during the early stages of Rift Valley Fever Virus (RVFV) infection.

The authors determine the crystal structure of Gn at atomic resolution and by combining available structures of Gc, Cryo-EM reconstruction and molecular dynamic they proposed a model for the assembly of the virus particle as well as suggested mechanism for the structural changes in the envelope preceding and during the membrane fusion step.

As the authors answered all the points from the previous round of review to my satisfaction. The manuscript, therefore, should be accepted for publication in nature communications.